# Facial Micro-Expression Recognition Enhanced by Score Fusion and a Hybrid Model from Convolutional LSTM and Vision Transformer

**DOI:** 10.3390/s23125650

**Published:** 2023-06-16

**Authors:** Yufeng Zheng, Erik Blasch

**Affiliations:** 1Department of Data Science, University of Mississippi Medical Center, Jackson, MS 39216, USA; 2MOVEJ Analytics, Fairborn, OH 45324, USA; erik.blasch@gmail.com

**Keywords:** facial micro-expression, human-machine interaction, long short-term memory (LSTM), convolutional neural network (CNN), vision transformer, score fusion, deep learning

## Abstract

In the billions of faces that are shaped by thousands of different cultures and ethnicities, one thing remains universal: the way emotions are expressed. To take the next step in human–machine interactions, a machine (e.g., a humanoid robot) must be able to clarify facial emotions. Allowing systems to recognize micro-expressions affords the machine a deeper dive into a person’s true feelings, which will take human emotion into account while making optimal decisions. For instance, these machines will be able to detect dangerous situations, alert caregivers to challenges, and provide appropriate responses. Micro-expressions are involuntary and transient facial expressions capable of revealing genuine emotions. We propose a new hybrid neural network (NN) model capable of micro-expression recognition in real-time applications. Several NN models are first compared in this study. Then, a hybrid NN model is created by combining a convolutional neural network (CNN), a recurrent neural network (RNN, e.g., long short-term memory (LSTM)), and a vision transformer. The CNN can extract spatial features (within a neighborhood of an image), whereas the LSTM can summarize temporal features. In addition, a transformer with an attention mechanism can capture sparse spatial relations residing in an image or between frames in a video clip. The inputs of the model are short facial videos, while the outputs are the micro-expressions recognized from the videos. The NN models are trained and tested with publicly available facial micro-expression datasets to recognize different micro-expressions (e.g., happiness, fear, anger, surprise, disgust, sadness). Score fusion and improvement metrics are also presented in our experiments. The results of our proposed models are compared with that of literature-reported methods tested on the same datasets. The proposed hybrid model performs the best, where score fusion can dramatically increase recognition performance.

## 1. Introduction

Facial expressions serve as a universally understood form of human communication intimately linked to one’s mental states, attitudes, and intentions. In addition to the typical facial expressions displayed in daily life, known as *macro-expressions*, there exists a distinct category called *micro-expressions*. These micro-expressions emerge in specific conditions, unveiling people’s concealed emotions during high-stakes situations when they strive to mask their true feelings [1,2]. Unlike macro-expressions, micro-expressions are involuntary, spontaneous, subtle, and rapid, lasting typically between 40 and 450 milliseconds. They are instinctive facial movements that react to emotional stimuli [3,4]. While individuals can consciously conceal or restrain their genuine emotions through macro-expressions, micro-expressions are beyond their control, inevitably exposing the authentic emotions they experience [5,6,7,8].

Recognizing micro-expressions is a daunting task as they are fleeting, involuntary, and exhibit low intensity. Only extensively trained experts possess the ability to discern these subtle facial cues. However, even with rigorous training, the average human can only identify approximately 47% of micro-expressions [9]. Moreover, human analysis of micro-expressions is error-prone and costly. Therefore, there is a pressing need to develop an automated system for the analysis and recognition of micro-expressions.

Facial expressions result from the intricate interplay of facial skin, connective tissue, and the activation of facial muscles controlled by facial nerve nuclei. These nuclei, in turn, are regulated by cortical and subcortical upper motor neuron circuits. A noteworthy neuropsychological study investigating facial expressions [10] unveiled two distinct neural pathways situated in different brain areas, each playing a role in mediating facial behavior. The cortical circuit, located within the cortical motor strip, primarily governs deliberate, voluntary facial expressions. Conversely, the subcortical circuit, residing in subcortical brain regions, primarily governs spontaneous, involuntary emotional facial expressions. In intense emotional situations where individuals endeavor to conceal or suppress their feelings, both systems are likely to be activated, resulting in fleeting glimpses of genuine emotions through micro-expressions [11]. Consequently, when attempting to mask their emotions, true feelings can swiftly “leak out” and manifest as micro-expressions [5].

Micro-expressions encompass the ability to convey seven universal emotions: disgust, anger, fear, sadness, happiness, surprise, and contempt [12]. Differing from macro-expressions, micro-expressions are characterized by their short duration and pronounced inhibition of facial muscle movement [1,13]. This distinctive feature allows them to authentically reflect a person’s true emotions, as they occur involuntarily and are more difficult to control [14]. However, micro-expression recognition is very challenging. According to a literature report [15] tested on the micro- and macro-expression warehouse (MMEW) dataset up to 2021, the best traditional machine learning (ML) method uses Directional Mean Optical Flow (MDMO) features [16], which can achieve 65.7% accuracy. Meanwhile, the best deep learning (DL) method applies a transferring long-term convolutional neural network (TLCNN) to extract frame features [17], which can be as high as 69.4% accuracy.

One of the challenges in micro-expression recognition is how to extract the temporospatial features from facial videos, which contain big and redundant data. Traditional feature extraction methods based on single image analysis [16] are not effective. Regular CNNs [17] are good for capturing spatial features but lack temporal analyses between sequential frames.

Therefore, in this study, to extract facial features from sequential frames (video clips), we propose a hybrid model comprised of a transformer and RNN (LSTM). The transformer can summarize the sparse spatial relations among image blocks [18], whereas the LSTM can analyze the temporal changes among frames. Furthermore, score fusion methods are applied to the multiple scores (from different NN models) in order to improve micro-expression recognition accuracies. The remainder of this paper is organized as follows. Section 2 describes datasets and preprocessing. Section 3 reviews the CNN and RNN. Section 4 presents the experimental results. Section 5 summarizes the paper.

## 2. Micro-Expression Dataset and Image Preprocessing

### 2.1. Facial Micro-Expression Dataset

There are two datasets used in this study, which are briefly described in this sub-section.

The MMEW dataset [15] follows the same elicitation paradigm used in other published datasets [19,20,21,22,23], i.e., watching emotional video episodes while attempting to maintain a neutral expression. The full details of the MMEW dataset construction process are presented in Reference [15]. All samples in MMEW were carefully calibrated by experts with onset, apex, and offset frames, and the action unit (AU) [15] annotation from the Facial Action Coding System (FACS) [24] was used to describe the facial muscle movement area. MMEW contains 300 micro-expression samples (image sequences). The samples in MMEW have a large image resolution (1920 × 1080 pixels). Furthermore, MMEW has a facial image size of 400 × 400 pixels. MMEW has seven elaborate emotion classes (see Figure 1), i.e., Happiness (36), Anger (8), Surprise (89), Disgust (72), Fear (16), Sadness (13), and Others (66).

The Chinese Academy of Sciences Micro-expression CASME II [22] dataset was developed in a well-controlled laboratory environment, where four lamps were chosen to provide steady and high-intensity illumination. To elicit micro-expressions, participants were instructed to maintain a neutral facial expression when watching video episodes with high emotional valence. CASME II used a high-speed camera with a sampling rate of 200 fps. There are 247 image sequences, which consist of facial images of 280 × 340 pixels. CASME II contains the samples of five emotion classes (see Figure 1), i.e., Happiness (33 samples), Repression (27), Surprise (25), Disgust (60), and Others (102).

### 2.2. Facial Image Standardization and Normalization

All facial images have been extracted from both datasets (as shown in Figure 1). Face detection is unnecessary in this study as the dataset contains one face per image.

General standardization is applied to all facial images, which is defined as follows.
(1)$$I_{N}^{\prime }=I_{N}-I_{M},$$
(2)$$I_{S}=\left(I_{N}^{\prime }-\mu \right)/\sigma ,$$
where **I**_N_ is the normalized facial image, **I**_M_ is the mean image of all faces in the dataset, and $I_{N}^{\prime }$ is their difference image. **I**_S_ is the standardized image; μ and σ denote the mean and standard deviation of the $I_{N}^{\prime }$ image, respectively. 

Image normalization (intensity scaling) is required by neural network models, which is defined as
(3)$$I_{N}^{}=(I_{0}^{}-I_{\mathrm{Min}}^{})\frac{L_{\mathrm{Max}}-L_{\mathrm{Min}}}{I_{\mathrm{Max}}^{}-I_{\mathrm{Min}}^{}}+L_{\mathrm{Min}},$$
where **I**_N_ is the normalized image, **I**_0_ is the original (input) image; *I*_Min_ and *I*_Max_ are the minimum and maximum pixel values in **I**_0_, while *L*_Min_ and *L*_Max_ are the desired minimum and maximum pixel values in **I**_N_. For example, we may select *L*_Min_ = 0 and *L*_Max_ = 1.0.

## 3. Convolutional and Recurrent Neural Networks

### 3.1. Convolutional Neural Networks

Convolutional neural networks (CNNs) draw inspiration from the biological functioning of the visual cortex, where small groups of cells exhibit sensitivity to specific regions within the visual field. CNNs blend principles from biology, mathematics, and computer science, making them pivotal innovations in the domains of computer vision and artificial intelligence (AI). The year 2012 marked a significant turning point for CNNs when Krizhevsky et al. [25] utilized an eight-layer CNN (comprising five convolutional layers and three fully-connected layers) to secure victory in the ImageNet competition. This groundbreaking achievement, known as *AlexNet*, reduced the classification error rate from 25.8% in 2011 to an impressive 16.4% in 2012, signifying a remarkable improvement at that time. Since then, deep learning CNNs have spurred the development of numerous applications in various domains.

During the training of AlexNet, batch *stochastic gradient descent* (SGD) was employed, incorporating carefully selected momentum and weight decay values. This groundbreaking model achieved exceptional performance on the challenging ImageNet dataset, setting a new record in the competition and solidifying the superior capabilities of CNNs. Later, there are many well-known CNNs developed, such as VGG-19, ResNet-50, Inception-V3, DenseNet-201, etc. Hereby our review begins with ResNet and Xception models, followed by a discussion on vision transformers, and then finishes with RNNs.

### 3.2. ResNet and Xception Models

The ResNet-50 model [26] is composed of 50 layers, featuring 16 residual blocks with three layers each, in addition to input and output layers. These *residual* blocks introduce identity connections that facilitate incremental or residual learning, enabling effective back-propagation. Through this approach, the identity layers progressively evolve from simple to complex representations. This evolution is particularly beneficial when the parameters of a CNN block start at or near zero. The inclusion of residual blocks helps address the challenging issue of vanishing gradients encountered in training deep neural networks with more than 30 layers.

Recently, an Xception [27] (Extreme Inception) network architecture has been proposed on the following hypothesis: the mapping of cross-channel correlations and spatial correlations in the feature maps of CNNs can be entirely decoupled. Thus, the Inception modules can be replaced with *depthwise separable convolutions*. The feature extraction base of the Xception architecture is constructed with 36 convolutional layers. For image classification, a logistic regression layer follows the convolutional base. Optionally, fully-connected layers can be added before the logistic regression layer. These 36 convolutional layers are organized into 14 modules, with linear residual connections encompassing each module, except for the first and last ones. When compared to Inception V3, Xception exhibits a comparable parameter count while demonstrating slight enhancements in classification performance on the ImageNet dataset.

In the Xception model, a depthwise separable convolution, also known as a separable convolution in deep learning frameworks, such as TensorFlow/Keras, is employed. This approach involves two steps: first, a depthwise convolution is performed independently on each channel of the input, followed by a pointwise convolution, which is a 1 × 1 convolution. The pointwise convolution projects the output channels from the depthwise convolution into a new channel space. The scenario of separable convolution plus pointwise convolution can significantly reduce the load of convolutional computation in contrast with a regular two-dimensional (2D) or three-dimensional (3D) convolutional layer; thus, it speeds up the CNN model training and the inference process.

### 3.3. Vision Transformers

Initially, transformers were developed and applied primarily to tasks in natural language processing (NLP), as evidenced by language models, such as BERT (Bidirectional Encoder Representations from Transformers) [28]. Transformers ascertain the connections between pairs of input tokens, such as words in NLP, through a mechanism called *attention*. However, this approach becomes increasingly computationally expensive with a growing number of tokens. When dealing with images, the fundamental unit of analysis becomes the pixel. Nevertheless, computing relationships between every pair of pixels becomes prohibitively costly in terms of memory and computation. To address this, *Vision Transformers* (ViTs) calculate relationships among smaller image regions, typically 16 × 16 pixels, resulting in reduced computational requirements. These regions, accompanied by positional embeddings, are organized into a sequence. The embeddings represent learnable vectors. Each region is vectorized and multiplied by an embedding matrix. The resulting sequence, along with the positional embeddings, is then fed into the transformer for further processing. The Video ViT (ViViT) model has one additional process, called a video tube (cube, i.e., frames by height by width, e.g., 4 × 16 × 16) positional embedding, while the rest of the ViViT process is the same as ViT.

Self-attention is commonly applied to the vision transformer model. The calculation of self-attention is to create three vectors from each of the encoder’s input vectors (in the NLP case, the embedding of each word). So for each word, a *Query* vector, a *Key* vector, and a *Value* vector are created by multiplying the embedding by three matrices that were trained during the training process. Multi-headed attention expands the model’s ability to focus on different positions. It gives the attention layer multiple “representation subspaces”. The multi-headed attention has multiple sets of Query/Key/Value weight matrices (e.g., a transformer uses eight attention heads, consisting of eight sets of weight matrices for each encoder/decoder). Each of these sets is randomly initialized. Then, after training, each set is used to project the input embeddings (or vectors from lower encoders/decoders) into a different representation subspace.

Two designs of the ViViT architecture are illustrated in Table 1, where the input is the frames of a video clip (shape of MMEW input: (14, 224, 224, 3)), while the output includes seven probabilities corresponding to seven micro-expression classes. The ViViT_FM2 model has an additional CNN block, X_CNN, which is comprised of the first five convolutional layers plus one residual block from the Xception model.

### 3.4. Recurrent Neural Networks—ConvLSTMmodels

Recurrent neural networks (RNNs) are designed to leverage sequential information in data. Unlike traditional neural networks that assume inputs and outputs are independent of each other, RNNs recognize the importance of dependencies in tasks such as natural language processing (NLP); for instance, predicting the next word in a sentence benefits from knowledge about the preceding words. RNNs are termed “recurrent” because they execute the same operation for each element of a sequence, with the output relying on previous computations. Another way to conceptualize RNNs is as having a “memory” that retains information about prior calculations. In theory, RNNs can utilize information from arbitrarily long sequences, but in practice, they are typically limited to considering only a few preceding steps. Figure 2 provides an illustration of a typical RNN architecture.

RNNs have exhibited remarkable achievements in various NLP tasks and applications involving temporal signals [29]. Among the different types of RNNs, *long short-term memory* (LSTM) networks are widely used and excel in capturing long-term dependencies. LSTMs are essentially similar to RNNs, but they employ a distinct method to compute the hidden state. In LSTMs, memories are referred to as cells, functioning, such as black boxes that take the previous (hidden) state, *s*_t−1_, and the current input, *x*_t_, as inputs. These cells internally make decisions about what information to retain or discard from memory. Subsequently, they combine the previous state, the current memory, and the input. Remarkably, these LSTM units have proven highly effective at capturing long-term dependencies.

In some applications (e.g., predicting weather changes), we want to model temporal evolution (e.g., temperature changing over time), ideally using recurrence relations (e.g., LSTM). In facial micro-expression recognition, we need to capture the facial muscle movement over time. At the same time, we also expect to efficiently extract spatial features (e.g., facial muscle movement varying with locations), something that is normally done with convolutional filters. Ideally, then, an architecture includes both recurrent and convolutional mechanisms, which are *convolutional LSTM* (ConvLSTM) layers.

As shown in Table 2, two designs of ConvLSTM models are illustrated for facial micro-expression recognition. The outputs of the 14 × 7 dimension correspond to 14 frames and seven micro-expression classes (MMEW), and the final recognition results of each video sample are the averaged results of 14 frames. The major difference between these two models is that the X_ConvLSTM_FM4 model is comprised of bidirectional ConvLSTM layers. In the context of an NLP model, a unidirectional ConvLSTM block can find some hints (such as the meaning of “it”) from future sentences, while a bidirectional ConvLSTM block can find hints from both future sentences and previous sentences. It is not surprising that the number of model parameters in the bidirectional X_ConvLSTM_FM3 model is more than doubled compared with that of a unidirectional X_ConvLSTM_FM4 model.

### 3.5. Hybrid Models

As described in previous subsections, we know that CNNs can extract spatial features; furthermore, ConvLSTM layers can capture spatial-temporal changes. CNNs are good at modeling neighborhood changes, whereas transformers with attention can grasp sparse spatial relations (e.g., among different image blocks or across frames in a video clip). As shown in Table 3, two designs of hybrid models combine three NN models: CNN, ConvLSTM, and ViViT. The two models differ in the ConvLSTM layer, where the Hybrid_FM5 model has 128 filters and is followed by a pooling layer. The goal is to combine the methods to enhance the performance of a single method.

The Hybrid_FM6 model is illustrated in Figure 3, which consists of three blocks: CNN, ConvLSTM, and Transformer. First, the CNN block (of six convolutional layers) provides local spatial features, where the feature image is the last-layer output randomly selected from 1 of 64 filters and 1 or 14 frames. Second, the ConvLSTM block (of one layer and 64 filters) generates temporal features, where the feature image is randomly selected from 1 of 64 filters and 1 or 14 frames. Third, the Transformer block (of 343 patches and 96 embedding dimensions) presents sparse spatial relations. Fourth, there is a fully connected layer (of 512 filters) prior to the output layer (of seven filters). The Hybrid_FM6 model can predict a micro-expression (e.g., “Anger”) with a given facial video (e.g., of 14 frames).

## 4. Experimental Results

Both datasets, MMEW and CASME II, were used in our experiments. The number of frames varies with different video files. Based on manual analyses of the minimal and maximal length of clips and filming speed (FPS), 14 frames are clipped in the MMEW dataset, whereas 24 frames are clipped in the CASME II dataset. If the number of frames in a video file is as *m* times long as the number of clipped frames (*n*), then *m* (typically *m* ≤ 2) samples (clips) are clipped from that file. However, there are no overlapped (repeatedly used) frames in the *m* samples from the same video file. The distributions of facial video samples (clips) from two datasets are shown in Figure 4, where the numbers of clips are larger than the number of video files. During the data splits (*k*-fold cross-validation) for training and testing, we will make sure that (i) multiple clips from one video file will split into one subset, i.e., either in training or in testing; (ii) the training set includes samples from all classes (stratified split).

All facial images were resized to 224 × 224 pixels, then standardized and normalized (intensity stretched). Ten-fold cross-validation was used in our experiments, and the final classification results were calculated by merging all 10-fold testing scores (instead of averaging 10 testing accuracies).

The first subsection briefly reviews the performance metrics used in our experiments. Then, the classification performances of eight NN models are presented. Score fusion improvement is described and quantitatively measured in the next two subsections. The time costs are reported in the last subsection.

### 4.1. Classification Performance Metrics

The performance of micro-expression recognition is measured by *F*1 score and accuracy, as shown in Table 4 and Table 5, where the two metrics are defined as follows.

The *Precision* is the ratio *TP*/(*TP* + *FP*) where *TP* is the number of true positives, and *FP is* the number of false positives. False positives are the samples that are predicted as positives but labeled as negatives. The *Precision* is intuitively the ability of the classifier not to predict a negative sample as positive.

The *Recall* is the ratio *TP*/(*TP* + *FN*) where *TP* is the number of true positives, and *FN* is the number of false negatives. False negatives are the samples that are predicted as negatives but labeled as positives. The *Recall* is intuitively the ability of the classifier to correctly predict all the positive samples.

The *F*1 score can be interpreted as a harmonic mean of the *Precision* and *Recall*, where an *F*1 score reaches its best value at 1 and worst score at 0. The relative contribution of *Precision* and *Recall* to the *F*1 score is equal. The formula for the *F*1 score is
*F*1 = 2 × (*Precision* × *Recall*)/(*Precision* + *Recall*).(4)
*Precision* = *TP*/(*TP* + *FP*)(5)
*Recall* = *TP*/(*TP* + *FN*)(6)
*Accuracy* = (*TP* + *TN*)/(*TP* + *FN* + *TN* + *FP*)(7)

In the multi-class and multi-label cases, the average of the *F*1 score of each class is analyzed with a weighting parameter. The weights for averaging can be calculated by the number of supported samples in each class divided by the total samples. The *F*1 score is an alternative to the *Accuracy* metric as it does not require one to know the total number of observations (e.g., *TN*). On the other hand, *Accuracy* tells how often we can expect a machine learning model will correctly predict an outcome out of the total number of predictions.

### 4.2. NN Model Performance on Two Datasets

Table 4 shows the F1 scores of seven micro-expressions, weighted F1 scores, and Accuracy values varying over eight different NN models when tested on the MMEW dataset. Based on the accuracy values, the model of Hybrid_FM6 achieves 0.8853, which is the best on the MMEW dataset. Compared to the literature-reported accuracy of 0.6940 on this dataset (by a CNN model in 2018—the best performance on the same dataset that we could find from the literature) [17], Hybrid_FM6’s accuracy is very high. It seems that ConvLSTM models are slightly better than ViViT models, both of which are better than CNN models (ResNet-50, Xception). According to the F1 scores, it looks as though “Anger” is the easiest to be recognized, while “Fear” and “Sadness” are mostly difficult to be detected.

Table 5 shows the F1 scores of seven micro-expressions, weighted F1 scores, and Accuracy values varying with eight different NN models when tested on the CASME II dataset. Based on the accuracy values, the model of Hybrid_FM5 is the best on the MMEW dataset, and its accuracy is 0.6565. Compared to the literature-reported accuracy of 0.6341 on this dataset (by an SVM classifier in 2014—the best performance on the same dataset that we could find from the literature) [29], Hybrid_FM5’s accuracy is pretty high. It seems that ConvLSTM models are slightly better than ViViT models, both of which are better than CNN models (ResNet-50, Xception). According to the F1 scores, it looks as though “Happiness” is the most difficult to be detected, while the other four expressions are equally hard to be recognized.

Overall, the hybrid models (combination of ResNet, ConvLSTM, ViViT) overperform non-hybrid models, such as CNN, ViViT, and ConvLSTM models.

### 4.3. Performance Improvement Using Score Fusion

The performance of facial micro-expression can be improved using score fusion methods, where the multiple scores are from eight different NN models, as presented in Table 4 and Table 5. There are several types of score-fusion methods: arithmetic fusion (e.g., average, majority vote) [30], classifier-based fusion, and density-based fusion (e.g., Gaussian mixture model) [31,32]. Based on score fusion performance [33,34], two classification-based score fusion methods are selected and presented in this study: Support-Vector Machine (SVM) and Random Forest (RF). The multiple scores are combined as feature vectors and then fed into a classifier for training (with labeled score vectors) or testing.

The Support-Vector Machine (SVM) is a supervised learning model utilized for non-linear classification and data analysis [35]. In the context of training data with categorized observations, the SVM training algorithm constructs a model that can assign new data points to specific categories. For classification purposes, an SVM establishes a hyperplane (or a set of hyperplanes) as a separating line between data points belonging to different classes. The objective is to find the optimal hyperplane that maximizes the distance between the hyperplane and the closest data points in each class. This approach effectively minimizes the generalization error of the classifier by maximizing the margin between the hyperplane and the nearest data points in each class [36].

Random forest (RF) is a classification model employed in supervised learning tasks. It leverages ensemble learning, which combines multiple models to tackle complex problems rather than relying on a single model. The RF algorithm enhances accuracy by utilizing bagging or bootstrap aggregating. It generates individual decision trees by using random subsets of the training dataset as subsamples. Each decision tree produces its own output or classification. The final output is determined through *majority voting*, where the RF output corresponds to the class chosen by the majority of trees. This approach effectively mitigates the impact of overfitting that may occur in individual decision trees. 

In this study, the SVM method employed a Gaussian kernel function and a one-versus-one coding design. This configuration resulted in the utilization of seven (or five) binary learners for the corresponding seven (or five) classes. In the RF model, we trained an ensemble comprising 100 classification trees using the complete training dataset. At each decision split, a random subset of predictors (scores) was utilized. The selection of split predictors aimed to maximize the gain of the split criterion across all possible splits of the predictors. The final classifications were obtained by combining the results from all the trees in the ensemble. 

Scores used for fusion are created in two ways: (i) a feature vector consists of eight class indices (e.g., 0, 1, … *n −* 1; *n* = 7 or 5) for the predicted classes (pred-class) from eight NN models; (ii) a feature vector consists of *n* accumulated probability values (sum-prob) per (onto) its predicted classes (*n* = 7 or 5) from eight different NN models. Three score feature-vector (pred-class or sum-prob) examples from datasets are shown in Table 6 and Table 7.

Table 6 lists six pred-class feature vectors (for six facial video samples in six rows), each of which consists of the predicted class index (0–6 for MMEW, 0–4 for CASME II) across eight NN models, where its probability value (between 0 and 1) is also given (to calculate sum-prob features). Table 7 presents lists six sum-prob feature vectors (in six columns), each of which consists of the accumulated probability values (sum-prob) with regard to its predicted classes across *n* micro-expresses (*n* = 7 or 5) for each facial video sample. For CASME II Sample 2, two models classified it as “Surprise” (1) with sum-prob = 1.4677, four models classified it as “Disgust” (2) with sum-prob = 2.2599, and the other two models classified it as “Others” (3) with sum-prob = 1.4687. The majority-voted result is “Disgust”.

In sum, each pred-class feature vector is the concatenated classification results (indices of classes) from different models, while each sum-prob feature vector is the summed classification probabilities from different models unfolded along with various micro-expressions.

The score fusion experiments are conducted using 10-fold cross-validation, and the final results (as shown in Table 6 and Table 7) are calculated with the merged 10-fold prediction outcomes. In the MMEW dataset, the RF method with pred-class features achieves the best overall. The accuracy of facial micro-expression recognition is improved to 0.9684 from 0.8853, which is a very good improvement. In the CASME II dataset, overall, the best method is still the RF method with pred-class features, which reaches 0.9112 in contrast with the best single NN model accuracy, 0.6565.

It seems the RF method with sum-prob features is better at recognizing some facial micro-expressions, such as Happiness, Surprise, and Disgust.

### 4.4. Metric for Fusion Improvement—Relative Rate Increase (RRI)

The performance improvement using score fusion (SF) cannot be properly measured by using the absolute difference of two accuracy rates (*R*_V_). For example, improving *R*_V_ from 80% to 90% seems to be more difficult than the improvement from 98% to 99%. Generally speaking, the improvement of *R*_V_ via SF becomes increasingly difficult when the original rate approaches 100%. Thus, it is proposed to use the Relative Rate Increase (RRI) [29] to evaluate the fusion improvement, where
(8)$$\mathrm{RRI}=\frac{\mathrm{ARI}}{1-\bar{R_{V}}}=\frac{R_{F}-\bar{R_{V}}}{1-\bar{R_{V}}},$$

*R*_F_ is the accuracy rate via SF and $\bar{R_{V}}$ is the mean of the accuracy rates from all classification models. If $\bar{R_{V}}$ = 1 (no need to improve the accuracy via SF), then set RRI = 1. The *absolute rate increase*
$\mathrm{ARI}=R_{F}-\bar{R_{V}},$ may not precisely measure the performance improvement as stated earlier. RRI ∈ (0, 1], where a higher value is better. According to the RRI definition, two fusion improvements—from 80% to 90% and from 98% to 99%—are equivalent, and both RRI values are 0.50. The two improvements are equivalent in the sense of their difficulty levels and the extent of the effort to implement them. Many metrics (e.g., F1, Precision, Recall) can be devised, wherein the RRI metric seeks to measure the actual improvement against the total amount of possible improvement. 

The RRI values from the best SF results are listed in the right-most columns in Table 8 and Table 9. In Table 8, RRI [F_1_(Anger)] = 1.0 means the SF improvement is perfectly done (cannot be better). RRI [F_1_(Fear)] = 0.8977 (the second best) means that the SF rate of 0.9708 is 89.77% improved in contrast with the mean rate of 0.7146. In Table 9, the best RRI [F_1_(Repression)] = 0.8914 represents that the SF rate of 0.9546 has an 89.14% improvement from the mean rate of 0.5821. The second best RRI [F_1_(Surprise)] = 0.8527 means an 85.27% SF increase on the basis of the averaged performance of individual models.

### 4.5. Time Costs of NN Models

All models were implemented with Tensorflow 2.10 and ran in Jupyterlab (Version 3.4.4) on a desktop computer, HP Omen, with the following configuration: Intel i7- 10700KF CPUs 3.8 GHz, 32 GB RAM, 1 TB hard disk, Ubuntu 20.04; NVIDIA GeForce RTX 3090 Graphics Board with 24 GB video memory (onboard) and 10,496 CUDA cores.

The number of model parameters and time costs of models are presented in Table 10 (on MMEW) and Table 11 (on CASME II). Time costs are related to the NN model (number of parameters) and data size (number of frames and samples). Model training is typically completed offline. In a real application, model inferencing (predicting) only processes one set of given frames or images. For example, using the Hybrid_FM6 model takes approx. 20 milliseconds per sample (14 frames) for predicting micro-expression, which is fast enough for real-time applications. The time costs on the CASME II dataset are longer due to processing more frame data (24 frames per video sample).

## 5. Summary and Discussion

In this study, we compared eight different neural network models in recognizing facial micro-expressions based on two datasets, where 6 of 8 models were newly designed for micro-expression recognition. The performance of the NN model in terms of accuracy (from high to low) is as follows: hybrid, ConvLSTM, vision transformer, and CNN. Overall we suggest the hybrid models that achieve the highest accuracy and yet are fast enough for real applications. The hybrid models are created by combining the fundamental building blocks from CNN, ConvLSTM, and vision transformer models, which are capable of extracting spatial features (in image neighborhood by CNN), summarizing temporal features (among video frames by LSTM), and capturing sparse spatial relations (among image blocks and video frames by transformer).

Score fusion can significantly increase facial micro-expression recognition rate. For example, “Fear” was only recognized at a low rate of 0.7146 (on the MMEW dataset). Random forest fusion improved the rate up to 0.9708, which is an 89.77% improvement according to the Relative Rate Increase (RRI) metric. The best overall accuracies from the hybrid models are 0.8853 (on the MMEW dataset) and 0.6565 (on the CASME II dataset), while score fusion can boost them up to 0.9684 and 0.9112. In addition, score fusion utilizes the outputs (e.g., predicted classes or probabilities) from multiple classifiers and has no additional hardware costs.

Information fusion can increase the recognition accuracy by combining the classification scores from different NN models and from different imaging modalities (e.g., infrared camera). With a large-scale dataset, the recognition reliability will also be improved. The inference latency may be further reduced with highly configured hardware (GPUs or multiple GPUs). 

Our experimental results shed light on a new method for real-time micro-expression recognition. Also, score fusion can further improve the recognition system performance without extra hardware costs. Real-time micro-expression recognition can be implemented and integrated into mobile devices or humanoid robots, which will enable a friendly human–machine interface taking micro-expressions into account for better decision-making.

## Figures and Tables

**Figure 1 sensors-23-05650-f001:**
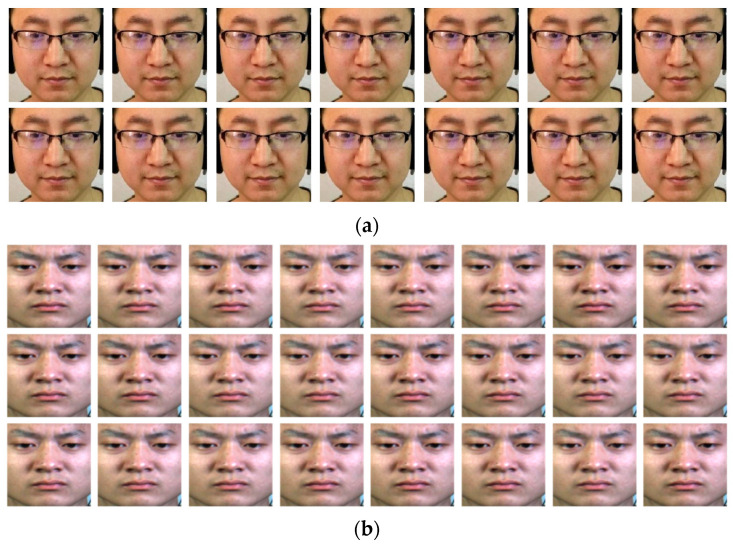
Facial micro-expression samples: (**a**) MMEW dataset—Facial video clip (14 frames shot at 90 FPS) labeled as “Happiness”; (**b**) CASME II dataset—Facial video samples (24 frames shot at 200 FPS) labeled as “Disgust”. All facial images are resized to 224 × 224 × 3.

**Figure 2 sensors-23-05650-f002:**
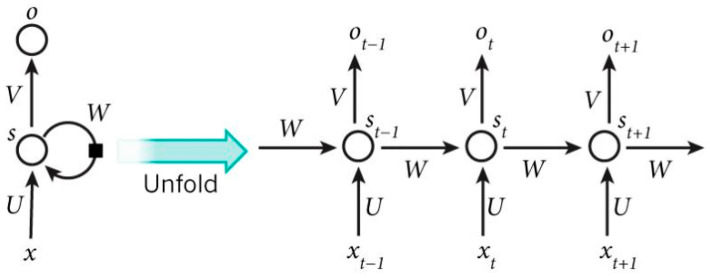
Illustration of a recurrent neural network and the unfolding in time of its forward computation.

**Figure 3 sensors-23-05650-f003:**
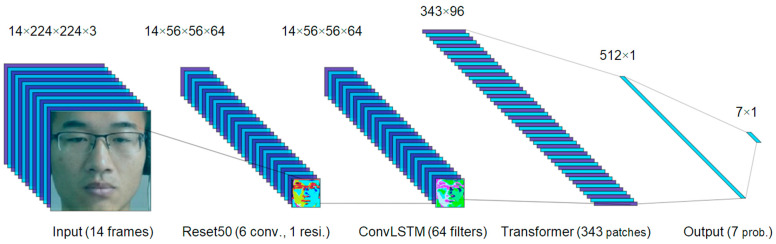
Feature maps of the Hybrid_FM6 model (refer to Table 3): The dimensions of feature maps are shown on the top, whereas the block functions are given at the bottom. The input is a video sample of 14 frames (the 1st MMEW sample), and the output is a vector of 7 probabilities: [0.0, 0.0, 0.0, 0.0, 0.0, 0.0, 1.0]. The index of the predicted class is 6 (max prob.), which corresponds to a labeled micro-expression, “Anger”.

**Figure 4 sensors-23-05650-f004:**
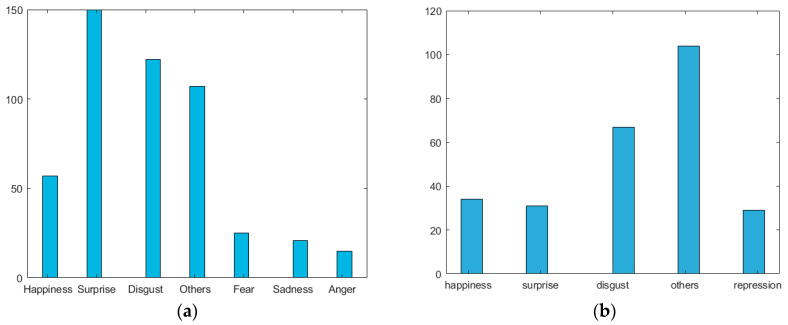
(**a**) Facial video clips (14-frame sequences @ 90 FPS) distributions of seven micro-expressions: a total of 497 samples from 300 original video clips in the MMEW dataset. (**b**) Facial video clips (24-frame sequences @ 200 FPS) distributions of five micro-expressions: a total of 265 samples from 247 original video clips in the CASME II dataset.

**Table 1 sensors-23-05650-t001:** Transformer model architectures—ViViT_FM1 and ViViT_FM2. Normalization and Dropout layers are omitted. The batch size (typically shown as None) is omitted in the “Output Shape” column. Attn_FF means attention-based feed-forward network. The numbers shown in “Output Shape” are assumed to be the inputs from the MMEW dataset.

ViViT_FM1 (8.6 M Paras)		ViViT_FM2 (8.8 M Paras)	
Layer (Type)	Output Shape	Layer (Type)	Output Shape
Frame_Input	(14, 224, 224, 3)	Frame_Input	(14, 224, 224, 3)
		time_distributed(X_CNN) (5 × Conv2D, 1 × Res.)	(14, 112, 112, 32)
Tubelet_Embedding (Conv3D → 7 × 16 × 16 patches)	(1792, 64)	Tubelet_Embedding (Conv3D → 7 × 16 × 16 patches)	(1792, 64)
Positional_Encoder	(1792, 64)	Positional_Encoder	(1792, 64)
**6 × Attn_FF:**		**6 × Attn_FF:**	
MultiHeadAttention (heads = 8, key_dim = 64)		MultiHeadAttention (heads = 8, key_dim = 64)	
Feed_Forward_Net (256 → 64)		Feed_Forward_Net (256 → 64)	
Add_Attn_FF_Norm	(1792, 64)	Add_Attn_FF_Norm	(1792, 64)
MaxPooling1D (pool_size = 4, strides = 4)	(448, 64)	MaxPooling1D (pool_size = 4, strides = 4)	(448, 64)
Flatten	(28672)	Flatten	(28672)
Dense	(256)	Dense	(256)
Dense (Output)	(7)	Dense (Output)	(7)

**Table 2 sensors-23-05650-t002:** RNN model architectures—X_ConvLSTM_FM3 and X_ConvLSTM_FM4. Normalization and Dropout layers are omitted. The batch size is omitted in the “Output Shape” column.time_distr = time_distributed. R_CNN consists of the first six convolutional layers plus one residual block from the ResNet-50 model.

X_ConvLSTM_FM3 (37.3 M Paras)		X_ConvLSTM_FM4 (17.5 M Paras)	
Layer (Type)	Output Shape	Layer (Type)	Output Shape
Frame_Input	(14, 224, 224, 3)	Frame_Input	(14, 224, 224, 3)
time_distr (R_CNN) (6 × Conv2D, 1 Res)	(14, 56, 56, 64)	time_distr (R_CNN) (6 × Conv2D, 1 Res)	(14, 56, 56, 64)
time_distr (MaxPool2D) (pool_size = (3,3), strides = (2,2))	(14, 28, 28, 64)		
Bidirectional_ConvLSTM2D (filters = 256, kernel_size = (3,3))	(14, 28, 28, 256)	ConvLSTM2D (filters=128, kernel_size=(3,3))	(14, 56, 56, 128)
time_distr (MaxPool2D) (pool_size = (2,2), strides = (2,2))	(14, 14, 14, 256)	time_distr (MaxPool2D) (pool_size = (2,2), strides = (2,2))	(14, 28, 28, 128)
Bidirectional_ConvLSTM2D (filters = 384, kernel_size = (3,3))	(14, 14, 14, 384)	ConvLSTM2D (filters = 256, kernel_size = (3,3))	(14, 28, 28, 256)
time_distr (MaxPool2D) (pool_size = (2,2), strides = (2,2))	(14, 7, 7, 384)	time_distr (MaxPool2D) (pool_size = (2,2), strides = (2,2))	(14, 14, 14, 256)
time_distr (Flatten)	(14, 18816)	time_distr (Flatten)	(14, 50176)
time_distr (Dense)	(14, 256)	time_distr (Dense)	(14, 256)
time_distr (Dense) (Output)	(14, 7)	time_distr (Dense) (Output)	(14, 7)

**Table 3 sensors-23-05650-t003:** Hybrid model architectures—Hybrid_FM5 and Hybrid_FM6 that combine 3 NN models: CNN, ConvLSTM, and ViViT. Normalization and Dropout layers are omitted. The batch size is omitted in the “Output Shape” column. time_distr = time_distributed.

Hybrid_FM5 (20.6 M Paras)		Hybrid_FM6 (20.4 M Paras)	
Layer (Type)	Output Shape	Layer (Type)	Output Shape
Frame_Input	(14, 224, 224, 3)	Frame_Input	(14, 224, 224, 3)
time_distr (R_CNN) (6 × Conv2D, 1 Res)	(14, 56, 56, 64)	time_distr (R_CNN) (6 × Conv2D, 1 Res)	(14, 56, 56, 64)
ConvLSTM2D (filters = 128, kernel_size = (3,3))	(14, 56, 56, 128)	ConvLSTM2D (filters = 64, kernel_size = (3,3))	(14, 56, 56, 64)
time_distr (MaxPool2D) (pool_size = (2,2), strides = (2,2))	(14, 28, 28, 128)		
Tubelet_Embedding (Conv3D → 7 × 7 × 7 patches)	(343, 96)	Tubelet_Embedding (Conv3D → 7 × 7 × 7 patches)	(343, 96)
Positional_Encoder	(343, 96)	Positional_Encoder	(343, 96)
**6 × Attn_FF:**		**6 × Attn_FF:**	
MultiHeadAttention (heads = 8, key_dim = 64)		MultiHeadAttention (heads = 8, key_dim = 64)	
Feed_Forward_Net (384 → 96)		Feed_Forward_Net (384 → 96)	
Add_Attn_FF_Norm	(343, 96)	Add_Attn_FF_Norm	(343, 96)
Flatten	(32928)	Flatten	(32928)
Dense	(512)	Dense	(512)
Dense (Output)	(7)	Dense (Output)	(7)

**Table 4 sensors-23-05650-t004:** F1 scores of seven micro-expressions, weighted *F*1 scores, and Accuracy values varying over eight different NN models tested on the MMEW dataset (14 frames at 90 FPS in each sample). The tests were conducted using 10-fold cross-validations. The prediction (probability) values from 10 validation folds were merged in one set to calculate the overall *F*1 (weighted average) and *Accuracy* scores. The highest *F*1 score or accuracy in each row is bolded. Notice that the highest accuracy of 0.6940 was reported on this dataset in 2018 [17].

Metric\NN Model	ResNet-50	Xception	ViViT_FM1	X_ViViT_FM2	X_ConvLSTM _FM3	X_ConvLSTM _FM4	Hybrid_FM5	Hybrid_FM6
*F*1 (Happiness)	0.8622	0.8935	0.9298	0.8947	0.8935	0.9273	0.9298	**0.9298**
*F*1 (Surprise)	0.9062	0.869	0.9067	0.8467	0.9100	0.8971	0.8667	**0.9133**
*F*1 (Disgust)	0.8601	0.8946	**0.9262**	0.9016	0.9104	0.9151	0.9098	0.9180
*F*1 (Others)	0.8284	0.8158	0.7850	0.7664	0.8178	0.8204	0.7944	**0.8318**
*F*1 (Fear)	0.7371	0.7086	0.6800	0.6400	0.7371	0.7343	**0.7600**	0.7200
*F*1 (Sadness)	0.7041	0.7551	**0.8095**	0.7143	0.7245	0.7381	0.7619	0.7619
*F*1 (Anger)	1.0	1.0	1.0	0.933	0.9952	0.9952	1.0	**1.0**
*F*1 (Weighted Avg.)	0.8601	0.8588	0.8760	0.8365	0.8753	0.8779	0.8675	**0.8855**
*Accuracy*	0.8588	0.8577	0.8752	0.835	0.8744	0.8765	0.8632	**0.8853**

**Table 5 sensors-23-05650-t005:** F1 scores of five micro-expressions, weighted *F*1 scores, and Accuracy values varying over eight different NN models tested on the CASME II dataset (24 frames at 200 FPS in each sample). The highest *F*1 score or accuracy in each row is bolded. Notice that the best accuracy of 0.6341 was reported in 2014 [29] on this dataset.

Metric\NN Model	ResNet-50	Xception	ViViT_FM1	X_ViViT_FM2	X_ConvLSTM _FM3	X_ConvLSTM _FM4	Hybrid_FM5	Hybrid_FM6
*F*1 (Happiness)	0.4545	0.4375	0.4706	0.3860	0.3980	0.4595	**0.5067**	0.5000
*F*1 (Surprise)	0.6809	0.6312	0.5882	0.6301	0.6288	0.6670	**0.6957**	0.6479
*F*1 (Disgust)	0.6398	0.6335	0.6466	0.5672	0.6364	0.6545	**0.6667**	0.6197
*F*1 (Others)	0.6910	0.7004	0.7014	0.6564	0.6695	0.7054	0.7058	**0.7281**
*F*1 (Repression)	0.5905	0.5299	0.6038	0.5588	0.5310	**0.6357**	0.6207	0.5862
*F*1 (Weighted Avg.)	0.6378	0.6325	0.6544	0.5934	0.6272	0.6627	**0.6739**	0.6543
*Accuracy*	0.6375	0.6238	0.6301	0.5886	0.6028	0.6468	**0.6565**	0.6452

**Table 6 sensors-23-05650-t006:** Three pred-class feature-vector examples from two datasets (top for MMEW and bottom for CASME II)—predicted class index (0–6 for MMEW, 0–4 for CASME II) and its probability value (between 0 and 1).

Dataset\Model	ResNet-50	Xception	ViViT_FM1	X_ViViT_FM2	X_ConvLSTM _FM3	X_ConvLSTM _FM4	Hybrid_FM5	Hybrid_FM6
MMEW Smpl 1	6 (0.9998)	6 (0.9117)	6 (0.9683)	6 (0.9598)	6 (0.9990)	6 (0.9906)	6 (0.9926)	6 (1.0000)
MMEW Smpl 2	2 (0.9988)	2 (0.8669)	4 (0.4080)	2 (0.5816)	2 (0.9956)	2 (0.9976)	2 (0.9678)	2 (1.0000)
MMEW Smpl 3	1 (0.9998)	1 (0.9317)	1 (0.5577)	4 (0.4170)	1 (0.5820)	1 (0.7767)	2 (0.2843)	2 (0.8847)
CASME II Smpl 1	3 (0.9892)	3 (0.5876)	3 (0.6981)	3 (0.4532)	3 (0.5273)	3 (0.8401)	3 (0.3726)	3 (0.8442)
CASME II Smpl 2	3 (0.9800)	1 (0.5690)	2 (0.3846)	2 (0.8025)	3 (0.4887)	1 (0.8988)	2 (0.4726)	2 (0.6003)
CASME II Smpl 3	2 (0.7515)	0 (0.4152)	0 (0.5328)	4 (0.8950)	2 (0.3420)	3 (0.4979)	4 (0.9895)	4 (0.8853)

**Table 7 sensors-23-05650-t007:** Three sum-probability feature-vector examples from two datasets (left for MMEW and right for CASME II). For the MMEW Sample 2, the ViViT_FM1 model classified it as “Fear” (4) with a probability of 0.4080 (refer to Table 6), while the rest of the seven models classified it as “Disgust” (2) with the accumulated probability value of 6.4083.

ME\Dataset	MMEW Smpl 1	MMEW Smpl 2	MMEW Smpl 3	CASME II Smpl 1	CASME II Smpl 2	CASME II Smpl 3	Dataset/ME
0 (Happiness)	0.0000	0.0000	0.0000	0.0000	0.0000	0.9480	0 (Happiness)
1 (Surprise)	0.0000	0.0000	3.8479	0.0000	1.4677	0.0000	1 (Surprise)
2 (Disgust)	0.0000	6.4083	1.1689	0.0000	2.2599	1.0936	2 (Disgust)
3 (Others)	0.0000	0.0000	0.0000	5.3123	1.4687	0.4979	3 (Others)
4 (Fear)	0.0000	0.4080	0.4170	0.0000	0.0000	2.7698	4 (Repression)
5 (Sadness)	0.0000	0.0000	0.0000	-	-	-	-
6 (Anger)	7.8217	0.0000	0.0000	-	-	-	-

**Table 8 sensors-23-05650-t008:** Improved performance via score fusion (SF)—*F*1 scores of seven micro-expressions, weighted *F*1 scores, and Accuracy values varying over two fusion methods vs. two combined scores originating from the MMEW dataset. The Mean Rate and Max Rate are computed (or extracted) from Table 4. The Best SF RRI values are calculated with the best SF rates (bolded, from Sum-Prob RF column and Pred-Class RF column) and the mean rates using Equation (8). The top two RRI values are highlighted with a shaded background.

Metric\NN Model	Mean Rate	Max Rate Hybrid_FM6	Pred-Class SVM	Pred-Class RF	Sum-Prob SVM	Sum-Prob RF	Best SF RRI
*F*1 (Happiness)	0.9076	0.9298	0.934	0.9739	0.9145	**0.9823**	0.8085
*F*1 (Surprise)	0.8895	0.9133	0.9103	0.9722	0.9126	**0.9743**	0.7675
*F*1 (Disgust)	0.9045	0.9180	0.9044	**0.9732**	0.9351	0.9691	0.7194
*F*1 (Others)	0.8075	0.8318	0.8774	**0.9619**	0.8637	0.9545	0.8021
*F*1 (Fear)	0.7146	0.7200	0.7750	**0.9708**	0.8336	0.9514	0.8977
*F*1 (Sadness)	0.7462	0.7619	0.7561	**0.9048**	0.8105	0.9017	0.6249
*F*1 (Anger)	0.9904	1.0	0.9952	**1.0**	1.0	0.9976	1.0000
*F*1 (Weighted Avg.)	0.8672	0.8855	0.8951	**0.9686**	0.9033	0.9664	0.7636
*Accuracy*	0.8658	0.8853	0.8948	**0.9684**	0.9024	0.9662	0.7646

**Table 9 sensors-23-05650-t009:** Improved performance via score fusion—*F*1 scores of seven micro-expressions, weighted *F*1 scores, and Accuracy values varying over two fusion methods vs. two combined scores originating from the CASME II dataset. The Mean Rate and Max Rate are computed (or extracted) from Table 5. The highest *F*1 score or accuracy in each row is bolded. The top two RRI values are highlighted with a shaded background.

Metric\NN Model	Mean Rate	Max Rate Hybrid_FM5	Pred-Class SVM	Pred-Class RF	Sum-Prob SVM	Sum-Prob RF	Best SF RRI
*F*1 (Happiness)	0.4516	0.5067	0.2338	0.8653	0.4681	**0.8726**	0.7677
*F*1 (Surprise)	0.6462	0.6957	0.5635	0.9377	0.7507	**0.9479**	0.8527
*F*1 (Disgust)	0.6331	0.6667	0.6041	0.8826	0.6689	**0.8871**	0.6923
*F*1 (Others)	0.6947	0.7058	0.7112	**0.9246**	0.7330	0.9161	0.7530
*F*1 (Repression)	0.5821	0.6207	0.6646	**0.9546**	0.6043	0.9383	0.8914
*F*1 (Weighted Avg.)	0.6420	0.6739	0.6290	**0.9143**	0.6707	0.9120	0.7606
*Accuracy*	0.6289	0.6565	0.6164	**0.9112**	0.6733	0.9093	0.7607

**Table 10 sensors-23-05650-t010:** Model parameters, training time (seconds per epoch), and testing time (milliseconds per sample) vary with NN models tested on the MMEW dataset. Time costs slightly change at different runs due to data caching and optimization.

Metric\NN Model	ResNet-50	Xception	ViViT_FM1	X_ViViT_FM2	X_ConvLSTM _FM3	X_ConvLSTM _FM4	Hybrid_FM5	Hybrid_FM6
Number of Parameters	25,693,063	22,966,831	33,934,215	13,029,487	28,597,319	17,578,183	20,643,719	20,446,727
Training Time (s/epoch)	17	22	17	101	34	22	15	17
Testing Time (ms/sample)	1	1	18	14	26	18	22	20

**Table 11 sensors-23-05650-t011:** Model parameters, training time (seconds per epoch), and testing time (milliseconds per sample) vary with NN models tested on the CASME II dataset. Time costs slightly change at different runs due to data caching and optimization.

Metric\NN Model	ResNet-50	Xception	ViViT_FM1	X_ViViT_FM2	X_ConvLSTM _FM3	X_ConvLSTM _FM4	Hybrid_FM5	Hybrid_FM6
Number of Parameters	25,691,013	22,964,781	29,823,877	13,569,133	28,596,805	17,577,669	18,603,941	18,800,165
Training Time (s/epoch)	17	21	8	165	32	21	14	19
Testing Time (ms/sample)	1	1	26	27	46	31	39	36

## Data Availability

Not applicable.

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
