# Peer review of "Facial Micro-Expression Recognition Enhanced by Score Fusion and a Hybrid Model from Convolutional LSTM and Vision Transformer"

_sensors, 2023, doi:10.3390/s23125650_

Round 1

Author Response

In this research, the authors investigated facial micro-expression recognition

enhanced by score fusion and a hybrid model from convolutional LSTM and

vision transformer. In order to improve this paper, the following issues shall

be considered:

  1. The main contribution of this paper shall be given in Introduction.

YZ: Our main contribution is “in this study, to extract facial features from sequential frames (video clips) we propose a hybrid model comprised of transformer and RNN (LSTM).” as revised in the Introduction section.

  1. The practical application of the derived results of this article shall be

given in Conclusion.

YZ: Thanks for the suggestion. The following sentence is added into the Conclusion.

The real-time micro-expression recognition can be implemented and integrated into mobile devices or humanoid robots, which will enable a friendly human-machine interface taking micro-expression into account for better decision making.

  1. The works shall be compared with the earlier works.

YZ: We compared our results with the best performance on the same dataset that we could find from the literature as described in the captions of Tables 4 & 5.

  1. What is the technical difficulty of the research methods in this paper?

YZ: As revised in the Introduction section, the technical difficulty is described as follows:

“One of the challenges in micro-expression recognition is how to extract the tem-porospatial features from facial videos, which contain big and redundant data. Tradi-tional feature extraction methods based on single image analysis [16] is not effective. Regular CNNs [17] are good to capture spatial feature but lack of temporal analyses between sequential frames.”

  1. The following works play a vital role in micro-expression recognition in

real-time applications. I suggest that the authors can add the following

works in this article.

YZ: Thank you for the suggestion. However, we cannot find any direct links between our work and the mentioned articles.

[1]Bifurcations in a fractional-order BAM neural network with four different

delays, Neural Networks 141 (2021) 344-354.

[2] New insight into bifurcation of fractional-order 4D neural networks incorporating

two different time delays, Communications in Nonlinear Science

and Numerical Simulation 118(2023) 107043.

[3]Local and global dynamics of a fractional-order predator-prey system with

habitat complexity and the corresponding discretized fractional-order system,

Journal of Applied Mathematics and Computing, volume 63, pages

311-340 (2020).

Reviewer 2 Report

Overall, the paper is well written and easy to follow.   Three NN modules are used but the sequence of use is not quite clear or convincing.   For example, the authors could have used a combination of CNN and transformers that is using multi-head attention on the CNN feature maps.  Fig. 3 the block diagram of the 3 NN-based modules is very poor. The authors need to provide a more detailed diagram. Some minor issues that the authors need to correct 

1. On page 4, ResNet50 has only 50-layer not 101 layer! It consists of 48 CNN layers. See the diagram in reference [24].

2. On page 7, Table 2 needs to insert space between X-ConvLSTM_FM and (37.3M parad).

3. On page 8 the word Subsection should be subsection.

4. On page 10 at the last sentence the word difficulty should be difficult .

Author Response

  1. On page 4, ResNet50 has only 50-layer not 101 layer! It consists of 48 CNN layers. See the diagram in reference [24].

YZ: Thank you for pointing out this mistake. It has been corrected in the revised manuscript.

  1. On page 7, Table 2 needs to insert space between X-ConvLSTM_FM and (37.3M parad).

YZ: Done.

  1. On page 8 the word Subsection should be subsection.

YZ: Corrected, Thank you.

  1. On page 10 at the last sentence the word difficulty should be difficult .

YZ: Revised, Thanks.

Reviewer 3 Report

1.  Article with a clear and correctly structured contribution.

2. The title and abstract reflect the content of the article.

3. It is advisable to add a section with an analysis of related work and present the differences with this work.

4. It is convenient to describe and justify the experimentation method applied from the literature.

Author Response

  1. Article with a clear and correctly structured contribution.

YZ: Thank you for your comments.

  1. The title and abstract reflect the content of the article.

YZ: Thanks.

  1. It is advisable to add a section with an analysis of related work and present the differences with this work.

YZ: The summary of related work is described in the Introduction section.

  1. It is convenient to describe and justify the experimentation method applied from the literature.

YZ: We compared our results with the best performance on the same dataset that we could find from the literature. The same note is added in the text.

Reviewer 4 Report

The problem tackled by the authors is relevant and interesting, and suitable to the journal. The overall technical contribution is minor, resulting from rather straightforward combination of well known techniques, but there is some value to it, if for no other reason than because of the results of comparing different methods.

The abstract should have the findings specified in more concrete terms. It is too vague now.

There are a number of technically misleading, at the very least, phrasings. The authors talk about score fusion, yet do no such thing. Score fusion refers to the use of multiple scores to arrive at a new score. What is done here instead is the use of multiple scores as features for classification, a completely different thing. This is why sentences such as:

"For classification purposes, an SVM establishes a hyperplane (or a set of hyperplanes) as a separating line between data points belonging to different classes."

are highly confusing to the reader. Only after reading the entire thing can the reader understand what the authors meant and then the section needs to be re-read with the new understanding in mind.

There are many linguistically poorly formed sentences such as:

"The hybrid models have the mixed architecture from CNN, ConvLSTM, and vision transformer..."

What is mixed about a CNN? The authors are not expressing properly what they mean to say.

Then there are odd/inappropriate word choices, such as:

"while the outputs are the micro-expressions gleaned from the videos."

Next, authors should always be referred to by their last name only and not, e.g., as:

"Alex Krizhevsky et al."

There are too many words/phrases which are emphasised, inconsistently too (sometimes in bold, sometimes italicized).

Lastly, important references to the most recent work in the area are missing, such as multiple papers by Zhang (e.g. "Short and long range relation based spatio-temporal transformer for micro-expression recognition", "Review of automatic microexpression recognition in the past decade")

As noted above, there are many linguistic issues which need to be corrected throughout the manuscript. I could not list even close to all, so proper care should be taken to revise the entire text in accordance with the suggestions I exemplified using a few problematic instances.

Author Response

The problem tackled by the authors is relevant and interesting, and suitable to the journal. The overall technical contribution is minor, resulting from rather straightforward combination of well known techniques, but there is some value to it, if for no other reason than because of the results of comparing different methods.

The abstract should have the findings specified in more concrete terms. It is too vague now.

YZ: Thank you for the suggestion. The abstract is revised.

There are a number of technically misleading, at the very least, phrasings. The authors talk about score fusion, yet do no such thing. Score fusion refers to the use of multiple scores to arrive at a new score. What is done here instead is the use of multiple scores as features for classification, a completely different thing. This is why sentences such as:

"For classification purposes, an SVM establishes a hyperplane (or a set of hyperplanes) as a separating line between data points belonging to different classes."

are highly confusing to the reader. Only after reading the entire thing can the reader understand what the authors meant and then the section needs to be re-read with the new understanding in mind.

YZ: Thank you for this constructive suggestion. The paragraph has been edited as follows:

“The performance of facial micro-expression can be improved using score fusion methods, where the multiple scores are from 8 different NN models as presented in Ta-bles 4 and 5. There are several types of score-fusion methods: arithmetic fusion (e.g., average, majority vote), classifier-based fusion, and density-based fusion (e.g., Gaussian mixture model). Based on score fusion performance [28], [29], two classification-based score fusion methods are selected and presented in this study: Support-Vector Machine (SVM) and Random Forest (RF). The multiple scores are combined as feature vectors then fed into a classifier for training (with labeled score vectors) or testing.”

There are many linguistically poorly formed sentences such as:

"The hybrid models have the mixed architecture from CNN, ConvLSTM, and vision transformer..."

What is mixed about a CNN? The authors are not expressing properly what they mean to say.

YZ: It is revised as “The hybrid models are created by combining the fundamental building blocks from CNN, ConvLSTM, and vision transformer models, …”

Then there are odd/inappropriate word choices, such as:

"while the outputs are the micro-expressions gleaned from the videos."

YZ: Changed to “… recognized …”

Next, authors should always be referred to by their last name only and not, e.g., as:

"Alex Krizhevsky et al."

YZ: You are correct. It is changed to “Krizhevsky et al."”

There are too many words/phrases which are emphasised, inconsistently too (sometimes in bold, sometimes italicized).

YZ: You are right. We checked all italicized words and changed many of them back to regular fonts.

Lastly, important references to the most recent work in the area are missing, such as multiple papers by Zhang (e.g. "Short and long range relation based spatio-temporal transformer for micro-expression recognition", "Review of automatic microexpression recognition in the past decade")

YZ: Thank you for the suggestions. The two papers were cited in the revised MS.

Round 2

Reviewer 1 Report

This paper can be accepted.